# Detection of an *Enterococcus faecium* Carrying a Double Copy of the *PoxtA* Gene from Freshwater River, Italy

**DOI:** 10.3390/antibiotics11111618

**Published:** 2022-11-13

**Authors:** Marzia Cinthi, Sonia Nina Coccitto, Gianluca Morroni, Gloria D’Achille, Andrea Brenciani, Eleonora Giovanetti

**Affiliations:** 1Department of Life and Environmental Sciences, Polytechnic University of Marche, 60121 Ancona, Italy; 2Department of Biomedical Sciences and Public Health, Polytechnic University of Marche, 60121 Ancona, Italy

**Keywords:** *Enterococcus faecium*, freshwater, oxazolidinones resistance, *poxtA*, conjugative plasmids, river

## Abstract

Oxazolidinones are valuable antimicrobials that are used to treat severe infections due to multidrug-resistant (MDR) Gram-positive bacteria. However, in recent years, a significant spread of clinically relevant linezolid-resistant human bacteria that is also present in animal and environmental settings has been detected and is a cause for concern. This study aimed to investigate the presence, genetic environments, and transferability of oxazolidinone resistance genes in enterococci from freshwater samples. A total of 10 samples were collected from a river in Central Italy. Florfenicol-resistant enterococci were screened for the presence of oxazolidinone resistance genes by PCR. *Enterococcus faecium* M1 was positive for the *poxtA* gene. The *poxtA* transfer (filter mating and aquaria microcosm assays), localization (S1-PFGE/hybridization), genetic context, and clonality of the isolate (WGS) were analyzed. Two *poxtA* copies were located on the 30,877-bp pEfM1, showing high-level identity and synteny to the pEfm-Ef3 from an *E. faecium* collected from an Italian coastal area. The isolate was able to transfer the *poxtA* to enterococcal recipients both in filter mating and aquaria microcosm assays. This is—to the best of our knowledge—the first detection of an enterococcus carrying a linezolid resistance gene from freshwater in Italy.

## 1. Introduction

Enterococci are members of the gut microbial communities of warm-blooded animals, including humans. Although regarded as commensals, they are also a common cause of human opportunistic infections worldwide and are responsible for urinary tract infections, bacteremia, infective endocarditis, and, rarely, intra-abdominal infections and meningitis [1]. The ESKAPE group (*Enterococcus faecium*, *Staphylococcus aureus*, *Klebsiella pneumoniae*, *Acinetobacter baumannii*, *Pseudomonas aeruginosa*, *Enterobacter* spp.) has been highlighted by the WHO as a rising cause of nosocomial and antibiotic-resistant infections in the last few decades, threatening public health [2].

Due to their ubiquity in animal feces and persistence in the environment, enterococci spread in many habitats, and they can be isolated from the soil, water, food of animal origin, sewage, and plants; therefore, they are valuable indicators of environmental fecal contamination [3].

The *Enterococcus* genus is known for its flexibility in responding to varying environmental conditions and for efficiently acquiring and exchanging antibiotic resistance genes [4].

The treatment of human infections due to MDR- and vancomycin-resistant enterococci (VRE) is limited to last-resort antibiotics: quinupristin/dalfopristin, oxazolidinones, and daptomycin [5].

Oxazolidinones, linezolid, and tedizolid are synthetic molecules that bind the V domain of the 23S rRNA of the 50S ribosomal subunit and that inhibit protein synthesis [6]. Despite its synthetic nature, linezolid resistance was first reported in enterococci in 2001, shortly after its introduction [7]. Although several surveillance studies have reported that the incidence of linezolid resistance was <1%, MDR enterococci have increasingly emerged, representing a significant risk to public health [8].

Resistance to linezolid can arise via mutations in the 23S rRNA and ribosomal proteins L3 and L4, but also through the acquisition of transferable resistance genes: *cfr* and its variants, *poxtA* and *poxtA2* as well as *optrA* [9]. Cfr and Cfr-like methylases confer resistance to five classes of antimicrobial agents, including phenicols, lincosamides, oxazolidinones, pleuromutilines, and streptogramin A (PhLOPS_A_ phenotype) via the post-transcriptional methylation of the 23S rRNA [10]. The ABC-F proteins OptrA and PoxtA lead to a decreased susceptibility to phenicols and oxazolidinones (including tedizolid) by a ribosomal protection mechanism [11].

Although oxazolidinones have only been approved for clinical use, an increasing number of linezolid-resistant enterococci has been detected in animal [9,12,13,14] and environmental settings [15,16,17].

The extensive use of florfenicol in farm animals can promote the spread of resistance to phenicols as well as to oxazolidinones as a result of co-selection mechanisms [12,18,19], and the association of florfenicol residues with the abundance of oxazolidinone resistance genes in livestock manures has been established [12]. Moreover, the florfenicol released from the feces of treated animals remains active in the soil, and it is able to exert selective pressure on environmental bacteria [20].

Italy has one of the highest consumptions of antimicrobials via food-producing animals in the EU, with significant effects in the emergence and spread of resistances [21]. The increasing demands for animal-derived foods has led to intensive and large-scale livestock breeding with the production of large quantities of animal manure employed as organic fertilizer in agriculture, which has been proven to be a potential hotspot for antibiotic resistance dissemination [22]. Manure-associated enterococci can be disseminated into water resources, representing an important source of contamination [23].

The purpose of this study was to investigate the occurrence, genetic environments, and transferability of the linezolid resistance genes in enterococci from freshwater samples.

To the best of our knowledge, this is the first report on an oxazolidinone resistance gene-harboring enterococcus isolated from a river in Italy.

## 2. Results

### 2.1. Detection of Oxazolidinone Resistance Genes in Florfenicol-Resistant Enterococci and Antimicrobial Susceptibility Profiles

Out of 10 total samples, only 2 were positive for the presence of florfenicol-resistant enterococci. Just one isolate, *E. faecium* M1, was found to be positive for the presence of the *poxtA* gene.

*E. faecium* M1 was resistant to florfenicol (64 mg/L), chloramphenicol (32 mg/L), erythromycin (>128 mg/L), and tetracycline (64 mg/L). However, the isolate was intermediate to linezolid (MIC, 4 mg/L) and susceptible to tedizolid (0.5 mg/L) and vancomycin (1 mg/L).

### 2.2. Location of PoxtA Gene and Detection of Circular Forms

Hybridization assays using a biotin-labelled *poxtA* probe occurred both on the chromosome and on five plasmids ranging in size from 4 to 87 kb.

Inverse PCR experiments indicated that the genetic environment of a circular form of *poxtA* was detectable.

The instability of the genetic environment of *poxtA* and its location on plasmids of different sizes and even on the chromosome suggest the intracellular mobility of the *poxtA*-carrying element due to IS-mediated recombination events.

### 2.3. Transferability in Filter Mating Experiments and in Aquaria Microcosm Assays

For each in vitro transfer experiment, three transconjugants were randomly selected and characterized to determine the presence of the *poxtA* gene and to determine their antibiotic susceptibility (Table 1).

In the filter mating experiments, *E. faecium* M1 was able to transfer the *poxtA* gene to *E. faecium* 64/3 and *E. faecium* Ef1 recipients with frequencies of 5.8 × 10^−2^ and 5 × 10^−3^, respectively. All of the transconjugants obtained using *E. faecium* 64/3 as a recipient exhibited resistance to florfenicol, chloramphenicol, tetracycline, and erythromycin; reduced susceptibility to linezolid; and susceptibility to tedizolid and vancomycin. When *E. faecium* Ef1 was used as a recipient, the transconjugants also exhibited resistance to linezolid (MIC, 8 mg/L) and tedizolid (MIC, 1 mg/L).

In aquaria microcosm assays, the *poxtA* gene was successfully transferred to the *E. faecium* 64/3 recipient only, with frequencies ranging from 5.0 × 10^−4^ to 2.9 × 10^−7^.

The transfer frequencies obtained in the filter mating experiments (using both recipients) were higher than those recorded at 24 h in the three aquaria microcosm assays. Overall, no significant differences in the transfer frequencies at 24 and 48 h between aquaria were recorded; at 96 h, *poxtA* gene transfer was never detected (Table 1).

### 2.4. WGS Analysis

The *E. faecium* M1 genome consisted of one chromosome (2,700,018 bp) and five plasmids (from 4.3 kb to 87 kb). ResFinder analysis of the genome revealed a complex resistome for the presence of several acquired antibiotic resistance genes in addition to *poxtA*: *fexB* (resistance to phenicols), *tet*(M), and *tet*(L) (resistance to tetracyclines); *msr*(C) (resistance to macrolides, lincosamides, and quinupristin streptogramins group B); *erm*(B) (resistance to macrolides, lincosamides, and streptogramins group B); *lnu*(B) (resistance to lincosamides); *lsa*(E) (resistance to lincosamides and dalfopristin streptogramins group A); and *aph*(3′)-III, *aac*(6′)-Ii, *ant*(6)-Ia, and *aac*(6′)-*aph*(2″) (resistance to aminoglycosides). All of these genes, except for *msr*(C) and *aac*(6′)-Ii, which showed a chromosomal location, were plasmid-associated.

No mutations were detected in the genes encoding the 23S rRNA or ribosomal proteins.

Virulome analysis displayed the presence of two acquired virulence genes: *acm* (encoding a collagen-binding adhesin) and *efaAfm* (encoding cell wall adhesins).

The search for virulence genes (*gelE*, *cylA*, *cylB*, *cylM*, *esp*, *ace*, *prgB*, *asa1*, and *ash701*) has been also been extended to the *E. faecium* Ef1 recipient through PCR (Appendix A). In the isolate, only the presence of the enterococcal surface protein Esp-encoding gene was demonstrated.

*E. faecium* M1 belonged to ST1036, which has been associated with human enterococci (https://pubmlst.org/bigsdb?db=pubmlst_efaecium_isolates, accessed on 20 June 2022).

WGS analysis indicated that the *poxtA* gene was located on a 30,877-bp plasmid (36% GC content) designated as pEfM1 (accession number ON009374) and containing 29 ORFs and belonging to the *rep2* replicon type (Figure 1).

pEfM1 had high-level identity (nucleotide identity 99%; cover 100%) and synteny to the conjugative 27-kb plasmid pEfm-Ef3 (accession number MT683615) detected in an *E. faecium* isolate from sediment collected in an Italian coastal area [15].

Both plasmids in the *poxtA* genetic environment, flanked by two IS*1216* elements in the same orientation, were located in a Tn*6657*-like transposon that also contained the *fexB* gene (accession number MH746818) [24]; the *tet*(L) and *tet*(M) genes, arranged in tandem, were located upstream of the Tn*6657*-like transposon (Figure 1). The comparative analysis of the two plasmids showed that in pEfM1, a second copy of the *poxtA* genetic environment was also inserted within the Tn*6657*-like upstream area of *orf8* (encoding the lipoprotein NPL/P60, Appendix A) (Figure 1).

It is well established that the instability of the *poxtA* genetic environment as well as its mobility within the same strain is related to the presence of the two IS*1216* that are responsible for transposition and translocation processes [12,25]; therefore, the IS*1216* copy upstream of the *orf8* in pEfm-Ef3 allowed the integration of the *poxtA* environment via homologous recombination. As a confirmation of this, the flanking regions of the *poxtA* genetic context showed no target duplication sites observed in transposition phenomena [26].

Although pEfM1 turned out to be transferable from *E. faecium* M1 to *E. faecium* 64/3 and *E. faecium* Ef1, its genetic backbone lacks a complete transfer machinery. However, plasmidome analysis of *E. faecium* M1 showed that a complete conjugation region (approximately 30 kb) was present in the 87-kb co-resident plasmid. In this region, a pilus assembly gene, *traG*, *virB4,* and *ltrC*-like topoisomerase–primase genes were in fact detected. Since it is well known that in enterococci, non-transferable plasmids can be mobilized, both in trans and in cis, through co-resident conjugative elements [27,28], pEfM1 could exploit the 87-kb plasmid for conjugative transfer.

## 3. Materials and Methods

### 3.1. Sampling, Sample Processing, and Enterococcal Isolation

Sampling activities were carried out in February 2021 in an area 2 km from the Salinello estuary, a river in Abruzzo region that flows into the Adriatic Sea. Near this river stretch, there is an urban area and a wastewater treatment plant.

Overall, 10 freshwater water samples of 500 mL each were collected throughout the month. Surface water sampling only took place when there was adequate flow at the sampling location such that water samples could be collected using a telescopic stem inserted into the 1 L sampling bottles in an upstream motion, midway between the surface and the stream bottom, without disturbing bottom sediment.

The samples were immediately delivered to the laboratory and filtered through 0.45 µm membranes (Merk Life Science, Milan, Italy). Then, the filters were vortexed, and the recovered cells were incubated in 30 mL of Azide broth (Oxoid, Basingstoke, UK) overnight at 37 °C. The next day, 200 µL of each enrichment culture were spread on Slanetz-Bartley agar plates (Oxoid, Basingstoke, UK) supplemented with florfenicol (10 mg/L). The inoculated plates were incubated at 37 °C for 48–72 h for the selection of florfenicol-resistant enterococci.

### 3.2. Genotypic and Phenotypic Characterization

Florfenicol-resistant enterococci were screened by PCR for the presence of *cfr* and its variants, the *optrA* and *poxtA* genes, using primer pairs, as previously described [29]. The PCR products were subjected to Sanger sequencing. Enterococci carrying linezolid resistance genes were tested for their susceptibility to florfenicol, chloramphenicol, linezolid, tetracycline, erythromycin, and vancomycin (SigmaAldrich, St. Louis, MI, USA) by standard broth microdilution assay, and resistance to tedizolid was tested using Etest strips (Liofilchem, Roseto degli Abruzzi, Italy). Susceptibility tests were interpreted according to CLSI breakpoints (CLSI M100-S27 document) [30]. *Enterococcus faecalis* ATCC 29212 was used as a quality control measure. Species identification was performed by MALDI-TOF (Vitek-MS, bioMérieux).

### 3.3. S1-PFGE, Southern Blotting, and Hybridization Assays

The total DNA, enclosed in agarose gel plugs, was digested with S1 nuclease (Thermo Fisher Scientific, Milan, Italy) and then separated by PFGE, as previously described [31]. After S1-PFGE, the DNA was blotted onto positively charged nylon membranes (Ambion-Celbio, Milan, Italy) and hybridized with suitable biotin-labelled DNA probes, as described elsewhere [32].

### 3.4. Detection of Circular Forms

In order to investigate the excision of the genetic environment, PCR mapping assays were performed using outward-directed primers (poxtA3 5-GACGAGCCGACCAACCACCT-3; poxtA4 5-TTGGATTTTTGTCCGCCTGAA-3) that were pair-targeting the linezolid resistance gene.

### 3.5. Mating Experiments

The conjugative transfer of linezolid resistance genes was assessed in both in vitro filter mating experiments [32] and in aquaria microcosm assays using *E. faecium* 64/3 [33] and *E. faecium* Ef1 as recipients. *E. faecium* Ef1, isolated from one of ten freshwater samples, was susceptible to florfenicol (MIC, 2 mg/L) but was resistant to rifampicin (MIC, 64 mg/L).

Transconjugants were selected on brain heart infusion agar (BHIA) (Oxoid, Basingstoke, UK) plates containing florfenicol (10 mg/L), fusidic acid (25 mg/L), and rifampicin (25 mg/L) when *E. faecium* 64/3 was used as the recipient and with florfenicol (10 mg/L) and rifampicin (16 mg/L) when the recipient was *E. faecium* Ef1.

In the aquaria microcosm assays, three different aerated aquariums containing 2 L of freshwater river (aquarium A), 2 L of osmotic water (aquarium B), and 2 L of osmotic water supplemented with florfenicol (0.05 mg/L) (aquarium C) were set up. At time zero, the donors and recipients were inoculated at the final concentrations of 4 × 10^7^ CFU/mL and 4.5 × 10^7^ CFU/mL, respectively. Then, at 24, 48, and 96 h, a water sample (200 mL) was taken from each aquarium and filtered. Cells that had been recovered and resuspended in 1 mL of sterile saline were plated onto selective BHIA plates (see above). Plates were incubated at 37 °C for 24 h and then examined for the presence of transconjugants.

Transconjugants were tested for the presence of linezolid resistance genes by PCR and for their susceptibility to florfenicol, chloramphenicol, linezolid, tedizolid, tetracycline, and erythromycin. The SmaI-PFGE assay was carried out to confirm the genetic background of the transconjugants by comparing their patterns with those of the donors and the relevant recipients. Conjugation frequencies were expressed as the ratio of the cell number (CFU/mL) of the transconjugants to the recipient.

### 3.6. WGS and Sequence Analysis

The genomic DNA was extracted by a QIAcube automated extractor using a DNeasy PowerLyzer PowerSoil Kit according to the manufacturer’s instructions (Qiagen, Hilden, Germany).

The extracted DNA was subjected to WGS by a hybrid approach using both a short-read Illumina MiSeq platform (MicrobesNG, Birmingham, UK) with 2 × 250 paired-end technology and a long-read sequencing approach (MinION, Oxford Nanopore Technologies, Oxford, UK). SPAdes 3.15.2 software was used for the hybrid assembly of short and long reads (http://bioinf.spbau.ru/spades, accessed on 20 June 2022). In silico identification of the acquired antimicrobial resistance genes, the ribosomal mutations involved in oxazolidinone resistance, the virulome, and clonal lineage were carried out using dedicated tools available at the Center for Genomic Epidemiology available at http://www.genomicepidemiology.org/, accessed on 20 June 2022 (MLST v.2.0, ResFinder v.3.2, VirulenceFinder 2.0, LRE-finder v.1.0, PlasmidFinder 2.1) and using BLAST suite (https://blast.ncbi.nlm.nih.gov/Blast.cgi, accessed on 20 June 2022 ).

### 3.7. Nucleotide Sequence Accession Numbers

The nucleotide sequence of the plasmid pEfM1 has been deposited in GenBank under accession number ON009374. The WGS data of the E. faecium M1 are available under the BioProject ID PRJNA850894 (accession no: JANCLZ000000000).

## 4. Discussion

Enterococci are opportunistic pathogens that are responsible for community-acquired and nosocomial infections. Two species, *E. faecium* and *E. faecalis*, are considered to be two of the most important nosocomial pathogens worldwide [34], but non-faecium and non-faecalis enterococci have been increasingly reported to cause human bloodstream, urinary tract, and surgical wound infections [35]. Therapeutic options to treat these infections are considerably limited by the increasing resistances to penicillin, aminoglycosides, and glycopeptides; oxazolidinones are among the few available last-resort antibiotics recommended for the treatment of severe infections caused by VRE and MDR enterococci [5].

Aquatic ecosystems harbor a vast pool of antibiotic resistance genes, and the horizontal gene transfer that takes place in water environments is considered one of the main sources of antibiotic resistance in the natural environment [36]. The presence of antibiotic-resistant bacteria in water environments is becoming particularly worrying in view of detecting resistant isolates that are critically important antibiotic classes for human medicine, including last-resort antimicrobials.

Major advances in understanding the phylogeny and the ecology of enterococci in environmental (secondary) habitats have been made over the past several decades [3]. *Enterococcus* species have frequently been described as carriers of antibiotic resistance across the One Health continuum. Hospital effluents, untreated sewage, and raw manure have been identified as the main hotspots for antibiotic-resistant enterococci and as sources for their environmental spread [37].

So far, only two studies have reported the occurrence of enterococci carrying linezolid resistance genes in water environments. Enterococci harboring linezolid resistance determinants have been detected in marine sediment and zooplankton from two Italian coastal areas [15], while *optrA*- and *poxtA*-carrying enterococci have recently been isolated from flowing surface water in Switzerland [16].

This is—to the best of our knowledge—the first detection of a *poxtA*-carrying *E. faecalis* isolate from river water in Italy. The MDR *E. faecium* isolate belonged to ST1036, which has been associated with human enterococci. The *poxtA* gene is duplicated on the 30,877-bp pEfM1 plasmid, showing high-level identity and synteny to the pEfm-Ef3 plasmid from an *E. faecium* isolate collected in an Italian coastal area [15].

Recently, the occurrence of two *poxtA* copies was reported in the genomes of two isolates of swine origin, *Lactobacillus salivarius* BNS11 and *E. hirae* HDC14-2. Nevertheless, the genes were not on the same genetic element but instead were on the pBNS11-37 plasmid and chromosome as well as on two different plasmids (pHDC14-2.27K and pHDC14-2.133K), respectively [9,38]. The significance of the two detected *poxtA* copies is unclear; the occurrence of a second gene does not appear to affect the susceptibility of *E. faecium* M1 to oxazolidinones since the isolate was intermediate to linezolid and susceptible to tedizolid.

In the in vitro transfer experiments, pEfM1 was successfully transferred to *E. faecium* recipients, even in the absence of complete transfer machinery. On the other hand, enterococcal non-conjugative plasmids can use the transfer system of a co-resident conjugal plasmid to move to recipients [28]. Interestingly, Shan et al. showed how mobilizable *poxtA*-carrying plasmids could transfer with the help of a conjugative plasmid by homologous recombination in *E. faecalis* and by replicative transposition in *Enterococcus lactis* [39]. pEfM1 may have used the 87-kb plasmid, in which a complete conjugation region is detectable, to move to recipients. Even if the results of our in vitro assays indicate that the *poxtA* gene transfer occurs at a high frequency, is necessary to consider that a large number of environmental factors, such as nanomaterials, various oxidants, and light, might affect the horizontal gene transfer in natural water environments [36].

It is increasingly evident that aquatic ecosystems could also serve as a reservoir of oxazolidinone resistance genes and contribute to the dissemination of linezolid-resistant enterococci, with serious effects on human health.

The presence of clinically relevant linezolid-resistant enterococci in water environments could reflect their spillover from human and/or animal reservoirs and could also indicate that freshwater rivers could represent a source of these resistance genes.

The effective control of oxazolidinones resistance requires a One Health approach to counteract the spread of linezolid-resistant bacteria and to prevent the development of environmental reservoirs of resistance genes that are transmissible to humans via different routes, including bathing, aquaculture, and fish product consumption.

## Figures and Tables

**Figure 1 antibiotics-11-01618-f001:**
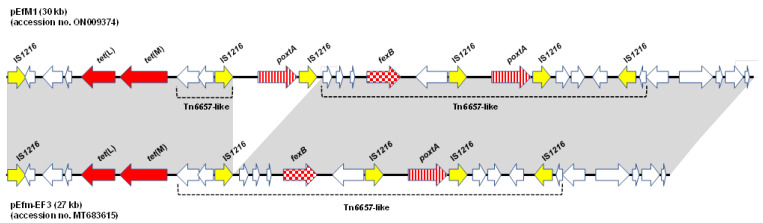
Schematic representation and comparison between the pEfM1 plasmid from *E. faecium* M1 (accession no. ON009374) and pEfm-EF3 plasmid (accession number MT683615).

**Table 1 antibiotics-11-01618-t001:** Frequencies of conjugation and antimicrobial susceptibility profiles for *poxtA*-carrying transconjugants.

		Recipient	Transfer Frequency	Transconjugants ^e^ MIC (mg/L) of: ^f^					
				FFC	CHL	LZD	TZD	TE	ERY
**Filter mating experiments**									
		*E. faecium* 64/3	5.8 × 10^−2^	64	32	4	0.5	128	128
		*E. faecium* Ef1	5 × 10^−3^	128	64	8	1	128	>128
**Aquaria microcosm assays**									
Aquarium A ^a^	24 h	*E. faecium* 64/3	6.3 × 10^−5^	64	32	4	0.5	128	>128
	48 h	*E. faecium* 64/3	1.8 × 10^−7^	64	32	4	0.5	128	>128
	96 h	*E. faecium* 64/3	ND ^d^						
Aquarium B ^b^	24 h	*E. faecium* 64/3	5 × 10^−4^	64	32	4	0.5	128	>128
	48 h	*E. faecium* 64/3	2.7 × 10^−7^	64	32	4	0.5	128	>128
	96 h	*E. faecium* 64/3	ND						
Aquarium C ^c^	24 h	*E. faecium* 64/3	9.2 × 10^−4^	64	32	4	0.5	128	>128
	48 h	*E. faecium* 64/3	2.9 × 10^−7^	64	32	4	0.5	128	>128
	96 h	*E. faecium* 64/3	ND						

^a^ Containing freshwater; ^b^ containing osmotic water; ^c^ containing osmotic water with florfenicol (0.05 mg/L); ^d^ not detectable. ^e^ Three transconjugants were characterized for each mating assay; their characteristics were overlapping, so only one transconjugant was shown in the table. ^f^ FFC, florfenicol; CHL, chloramphenicol; LZD, linezolid; TZD, tedizolid; TE, tetracycline; ERY, erythromycin; VAN, vancomycin.

## Data Availability

The data presented in this study are available on request from thecorresponding author.

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
