# Peer review of "Detection of an Enterococcus faecium Carrying a Double Copy of the PoxtA Gene from Freshwater River, Italy"

_antibiotics, 2022, doi:10.3390/antibiotics11111618_

Round 1
Reviewer 1 Report
Cinthi et al present the first report from Italy on oxazolidinone resistance gene-harboring Enterococcus faecium. The study adds to the lines of evidence that highlight the aquatic environment as an important reservoir of antimicrobial resistant organisms. The manuscript is well written, and overall, is an important contribution to the field of antimicrobial resistance. Nonetheless, there are some issues with grammar and clarity, which the authors need to address:
Line 11: Please rewrite “antimicrobials to treat” as “antimicrobials used to treat”.
Line 12: Please rewrite “clinically relevant” as “clinically-relevant”.
Line 14: Please rewrite “causes concern” as “is a cause for concern” and “This study aims” as “This study aimed”.
Line 40: Please rewrite “Enterococcus genus” as “The Enterococcus genus”.
Line 45: Please delete the “the” in “the protein synthesis”.
Line 46: Please rewrite “in 23S rRNA” as “in the 23S rRNA”.
Line 50: Please rewrite “a post-transcriptional” as “post-transcriptional”.
Line 54: Please rewrite “despite oxazolidinones” as “although oxazolidinones”.
Lines 57 to 60: Please rewrite “the spread of the resistance” as “the spread of resistance”. Also, please break the sentence into two, with the first sentence ending in the citation “[10,16,17]”. The second part of the sentence should stand on its own, and should be further revised to improve clarity, as it is difficult to understand what is being communicated.
Line 62: Please delete “also”.
Lines 65 to 66: Please rewrite “this is the first report on an enterococcus from river harboring an oxazolidinone resistance gene, in Italy” as “In Italy, this is the first report on an oxazolidinone resistance gene-harboring enterococcus isolated from a river” to improve the clarity of the sentence.
Lines 71 to 72: Please rewrite “of poxtA gene” as “of the poxtA gene”.
Lines 74 to 75: Please rewrite “the isolate was intermediate to linezolid” as “the isolate was intermediately resistant to linezolid”.
Lines 79: Please rewrite “on chromosome” as “on the chromosome”.
Line 82: Please rewrite “suggested the” as “suggest an”, and delete the “an” before “intracellular”.
Lines 96 to 97: Please rewrite “ranged” as “ranging”.
Lines 101: Please rewrite “have been” as “were”.
Lines 111: Please rewrite “of genome” as “of the genome”.
Lines 167: Please rewrite “Near to this river” as “Near this river”.
Lines 168: Please rewrite “treatment plan” as “treatment plant”.
Lines 169: Please rewrite “have been” as “were”.
Lines 170: Please rewrite “0,45” as “0.45”.
Lines 171: Please change the position of “overnight”, so that it occurs after “broth”.
Lines 177: Please introduce “as” in front of “previously”.
Lines 182: Please rewrite “CLSI breakpoint” as “CLSI breakpoints”.
Lines 189: Please rewrite “DNA” as “the DNA” and “positively charged” as “positively-charged”.
Lines 194: Please rewrite “assay” as “assays”.
Lines 199: Please rewrite “in vitro” as “in vitro”.
Lines 203: Please provide further details about “Oxoid”.
Lines 209: Please rewrite “have been set up” as “were set up”.
Lines 248: Please rewrite “recently been” as “been recently”.
Lines 250: Please rewrite “oxazolidinones resistance genes” as “oxazolidinone resistance genes”.
Author Response
Cinthi et al present the first report from Italy on oxazolidinone resistance gene-harboring Enterococcus faecium. The study adds to the lines of evidence that highlight the aquatic environment as an important reservoir of antimicrobial resistant organisms. The manuscript is well written, and overall, is an important contribution to the field of antimicrobial resistance. Nonetheless, there are some issues with grammar and clarity, which the authors need to address:
Response: We thank the reviewer for his/her positive comments.
Line 11: Please rewrite “antimicrobials to treat” as “antimicrobials used to treat”.
Response: Done
Line 12: Please rewrite “clinically relevant” as “clinically-relevant”.
Response: Done
Line 14: Please rewrite “causes concern” as “is a cause for concern” and “This study aims” as “This study aimed”.
Response: Done
Line 40: Please rewrite “Enterococcus genus” as “The Enterococcus genus”.
Response: Done
Line 45: Please delete the “the” in “the protein synthesis”.
Response:Done
Line 46: Please rewrite “in 23S rRNA” as “in the 23S rRNA”.
Response:Done
Line 50: Please rewrite “a post-transcriptional” as “post-transcriptional”.
Response: Done
Line 54: Please rewrite “despite oxazolidinones” as “although oxazolidinones”.
Response: Done
Lines 57 to 60: Please rewrite “the spread of the resistance” as “the spread of resistance”. Also, please break the sentence into two, with the first sentence ending in the citation “[10,16,17]”. The second part of the sentence should stand on its own, and should be further revised to improve clarity, as it is difficult to understand what is being communicated.
Response: Done
Line 62: Please delete “also”.
Response:Done
Lines 65 to 66: Please rewrite “this is the first report on an enterococcus from river harboring an oxazolidinone resistance gene, in Italy” as “In Italy, this is the first report on an oxazolidinone resistance gene-harboring enterococcus isolated from a river” to improve the clarity of the sentence.
Response: Done
Lines 71 to 72: Please rewrite “of poxtA gene” as “of the poxtA gene”.
Response:Done
Lines 74 to 75: Please rewrite “the isolate was intermediate to linezolid” as “the isolate was intermediately resistant to linezolid”.
Response: We believe that the term “intermediate” can be equally correct.
Lines 79: Please rewrite “on chromosome” as “on the chromosome”.
Response: Done
Line 82: Please rewrite “suggested the” as “suggest an”, and delete the “an” before “intracellular”. Done
Lines 96 to 97: Please rewrite “ranged” as “ranging”.
Response: Done
Lines 101: Please rewrite “have been” as “were”.
Response: Done
Lines 111: Please rewrite “of genome” as “of the genome”.
Response: Done
Lines 167: Please rewrite “Near to this river” as “Near this river”.
Response: Done
Lines 168: Please rewrite “treatment plan” as “treatment plant”. Done
Lines 169: Please rewrite “have been” as “were”.
Response: Done
Lines 170: Please rewrite “0,45” as “0.45”.
Response: Done
Lines 171: Please change the position of “overnight”, so that it occurs after “broth”.
Response: Done
Lines 177: Please introduce “as” in front of “previously”.
Response: Done
Lines 182: Please rewrite “CLSI breakpoint” as “CLSI breakpoints”.
Response: Done
Lines 189: Please rewrite “DNA” as “the DNA” and “positively charged” as “positively-charged”.
Response: Done
Lines 194: Please rewrite “assay” as “assays”.
Response: Done
Lines 199: Please rewrite “in vitro” as “in vitro”.
Response: Done
Lines 203: Please provide further details about “Oxoid”.
Response: Done
Lines 209: Please rewrite “have been set up” as “were set up”.
Response: Done
Lines 248: Please rewrite “recently been” as “been recently”.
Response: Done
Lines 250: Please rewrite “oxazolidinones resistance genes” as “oxazolidinone resistance genes”.
Response: Done
Reviewer 2 Report
Interesting work. I recommend a review on the pages layout
Author Response
Interesting work. I recommend a review on the pages layout
Response: We thank the reviewer for his/her positive comments.
Reviewer 3 Report
The manuscript discusses many interesting facts about the genetic backbone of Enterococcus faecium carrying linezolid resistance gene (poxtA gene ) . The manuscript is scientifically sound and well written.
Author Response
The manuscript discusses many interesting facts about the genetic backbone of Enterococcus faecium carrying linezolid resistance gene (poxtA gene ) . The manuscript is scientifically sound and well written.
Response: We thank the reviewer for his/her positive comments.
Reviewer 4 Report
The manuscript titled ‘Detection of an Enterococcus faecium carrying a double copy of the poxtA gene from freshwater river, Italy’ aimed to investigate the presence, genetic environments and transferability of oxazolidinone resistance genes in enterococci from freshwater samples. However, this study seems like a survey using a too low quantity of environmental samples. The study should be a part of a more comprehensive scientific concept which should be present in a common manuscript. Additionally, the Discussion is too limited.
Author Response
The manuscript titled ‘Detection of an Enterococcus faecium carrying a double copy of the poxtA gene from freshwater river, Italy’ aimed to investigate the presence, genetic environments and transferability of oxazolidinone resistance genes in enterococci from freshwater samples. However, this study seems like a survey using a too low quantity of environmental samples. The study should be a part of a more comprehensive scientific concept which should be present in a common manuscript. Additionally, the Discussion is too limited.
Response: Although we agree with the reviewer about the low number of environmental samples, our paper aimed to report of the first detection in Italy of an Enterococcus faecium carrying an oxazolidinone resistance gene from freshwater river rather than provide a surveillance on linezolid resistance in environment. The paragraph "Results and Discussion" has now been splitted into two separate sections "Results" and "Discussion" in the revised manuscript. Furthermore, the discussion has been extended as suggested.
Reviewer 5 Report
The studies related to the poxtA gene in enterococci have intensified in the last 3 years. This makes the presented manuscript up-to-date, but the presentation of the results and their discussion should be improved. The discussion should be expanded and deepened by presenting relevant references.
I have the following recommendations:
1. It is good to add the word "river" as a keyword (in connection with a more comprehensive search).
2. In "Abstract", line 13 - add the “humans”. In the main text, also discuss the case "humans" (example: lines 54-55), because there are a number of publications from the last 2-3 years related to detected poxtA gene in enterococci (E. faecium) from patients. This will describe the entire chain of transmission and movement of the poxtA gene of E. faecium - environment, animals and people.
3.The "Introduction" should end with the aims and objectives of the study. In this regard, the text between lines 63-66 should be corrected.
4.MM: The results of the applied methods to be presented in a figure/s. There are only 2 isolates, of which only one is described, because of that it will be better the authors to investigate pathogenicity and virulence factors of the isolates.
Lines 166-173: The sampling is a critical point and therefore the procedure or the protocol must be described.
Lines 194-196: add a citation if primers are not original.
The applied methods and the results allow to have two separated sections "Results" and "Discussion". This will improve the quality of presentation and it will allow some conclusions to be explained (eg lines 82-83; 124-125 - discuss the sequence type). There is currently no discussion.
New and actual methods have been applied and therefore it is good to present additional figures from the obtained results.
The results from table 1 should be presented in another way (as a figure, diagram or different table). The size of the table does not correspond to the volume of the data.
The sequencing results (“Genotypic and phenotypic characterization”) to be presented and discussed.
5. Conclusions: "Conclusions" to be adapted according to the extended discussion. In "Discussion" discuss the significance and role of the detected "two poxtA copies" (line 244) with plasmid and chromosomal localization.
Author Response
The studies related to the poxtA gene in enterococci have intensified in the last 3 years. This makes the presented manuscript up-to-date, but the presentation of the results and their discussion should be improved. The discussion should be expanded and deepened by presenting relevant references.
Response: We thank the reviewer for his/her positive comments. The discussion has been extended as suggested and carefully referenced.
1.The "Introduction" should end with the aims and objectives of the study. In this regard, the text between lines 63-66 should be corrected.
Response: The introduction has been modified as suggested (lines 76-79).
2.The results of the applied methods to be presented in a figure/s. There are only 2 isolates, of which only one is described, because of that it will be better the authors to investigate pathogenicity and virulence factors of the isolates
Response: We take note of the reviewer's suggestion, however we believe that figures could lead the reader out of the topic. The search of virulence genes has been also extended to E. faecium Ef1 recipient as suggested (lines 140-142).
3.Lines 166-173: The sampling is a critical point and therefore the procedure or the protocol must be described
Response: The paragraph “Sampling, sample processing and enterococcal isolation” has been expanded by adding more detailed information as suggested (lines 182-196)
4.The applied methods and the results allow to have two separated sections "Results" and "Discussion". This will improve the quality of presentation and it will allow some conclusions to be explained (eg lines 82-83; 124-125 - discuss the sequence type). There is currently no discussion.
Response: The Results and Discussion section has now been separated in two specific paragraph in the revised manuscript. The discussion has been expanded as suggested.
5.New and actual methods have been applied and therefore it is good to present additional figures from the obtained results.
Response: please see comment at point 2.
6.The results from table 1 should be presented in another way (as a figure, diagram or different table). The size of the table does not correspond to the volume of the data.
Response: The Table 1 has been modified as suggested.
7.The sequencing results (“Genotypic and phenotypic characterization”) to be presented and discussed.
Response: The plasmid sequence has already been provided during the submission (please see supplementary materials not to be published) and the discussion has been extended as suggested.
8.Conclusions: "Conclusions" to be adapted according to the extended discussion. In "Discussion" discuss the significance and role of the detected "two poxtA copies" (line 244) with plasmid and chromosomal localization.
Response: The “Conclusions” section has now been moved in the “Discussion” paragraph and the hypothetical role of the double copy of poxtA was discussed (Lines 299-301).
Round 2
Reviewer 4 Report
The manuscript has been improved.